# Three-Stage Single-Chambered Microbial Fuel Cell Biosensor Inoculated with *Exiguobacterium aestuarii* YC211 for Continuous Chromium (VI) Measurement

**DOI:** 10.3390/s19061418

**Published:** 2019-03-22

**Authors:** Li-Chun Wu, Guey-Horng Wang, Teh-Hua Tsai, Shih-Yu Lo, Chiu-Yu Cheng, Ying-Chien Chung

**Affiliations:** 1Department of Logistics Engineering, Dongguan Polytechnic, Dongguan City 523808, China; mic.wu@msa.hinet.net; 2Research Center of Natural Cosmeceuticals Engineering, Xiamen Medical College, Xiamen 361008, China; wanggh@livemail.tw; 3Department of Chemical Engineering and Biotechnology, National Taipei University of Technology, Taipei 10608, Taiwan; thtsai@ntut.edu.tw; 4Department of Biological Science and Technology, China University of Science and Technology, Taipei 11581, Taiwan; HBO223080284@gmail.com (S.-Y.L.); cycheng@cc.cust.edu.tw (C.-Y.C.)

**Keywords:** biosensor, chromium, microbial fuel cell, wastewater

## Abstract

Chromium (VI) [Cr(VI)] compounds display high toxic, mutagenic, and carcinogenic potential. Biological analysis techniques (e.g., such as enzyme-based or cell-based sensors) have been developed to measure Cr(VI); however, these biological elements are sensitive to the environment, limited to measuring trace Cr(VI), and require deployment offsite. In this study, a three-stage single-chambered microbial fuel cell (SCMFC) biosensor inoculated with *Exiguobacterium aestuarii* YC211 was developed for in situ, real-time, and continuous Cr(VI) measurement. A negative linear relationship was observed between the Cr(VI) concentration (5–30 mg/L) and the voltage output using an SCMFC at 2-min liquid retention time. The theoretical Cr(VI) measurement range of the system could be extended to 5–90 mg/L by connecting three separate SCMFCs in series. The three-stage SCMFC biosensor could accurately measure Cr(VI) concentrations in actual tannery wastewater with low deviations (<7%). After treating the wastewater with the SCMFC, the original inoculated *E. aestuarii* remained dominant (>92.5%), according to the next-generation sequencing analysis. The stable bacterial community present in the SCMFC favored the reliable performance of the SCMFC biosensor. Thus, the three-stage SCMFC biosensor has potential as an early warning device with wide dynamic range for in situ, real-time, and continuous Cr(VI) measurement of tannery wastewater.

## 1. Introduction

Hexavalent chromium [Cr(VI)] is a common environmental pollutant, used extensively in numerous industrial processes (e.g., tanning, electroplating, wood preservation, textile dyeing, and ore refining) [1]. Because of its high oxidizing potential, Cr(VI) easily causes toxic, mutagenic, and carcinogenic effects on biological organisms and has been identified as one of the 17 chemicals posing the greatest threat to humans by the US Environmental Protection Agency [2]. Thus, real-time monitoring and bioremediation of Cr(VI) are essential to protect human health and the environment.

To date, various bacterial strains, including *Pseudomonas* sp., *Enterobacter aerogenes*, *Serratia proteamaculans*, *Bacillus* sp., *Microbacterium* sp., *Trichococcus pasteurii*, *Desulfovibrio vulgaris*, *Ochrobactrum* sp., *Escherichia coli*, *Shewanella algae*, *Paenibacillus ferrarius*, *Exiguobacterium aestuarii*, and *Stenotrophomonas maltophilia*, capable of reducing Cr(VI) to Cr(III), have been isolated from various environments under anaerobic or aerobic conditions [3,4,5,6,7,8,9]. Although most of these microbes have been isolated, the availability of high selectivity, high reducing power, broad environmental tolerance, and anaerobic Cr(VI)-reducing bacteria is a prerequisite for accurate Cr(VI) measurement from actual wastewater using microbial fuel cells (MFCs).

Numerous analysis methods have been developed for Cr(VI) measurement in water samples. Commonly used chemical analysis techniques include atomic absorption spectrometry, inductively coupled plasma mass spectroscopy, ion chromatography, and colorimetric methods based on diphenylcarbazide [10]. These methodologies are effective and sensitive and have low detection limits; however, complicated operating procedures, expensive equipment, and long measurement times often restrict their application, especially for in situ or real-time Cr(VI) measurement [11]. Recently, biological analysis techniques have been considered for Cr(VI) measurement. They can compete with the chemical methods because of their simpler operation, cheaper equipment, and shorter measurement periods [7]. These biological analysis techniques include enzyme-based sensors (e.g., those using urease or cytochrome c3) and cell-based sensors (e.g., those using V79 cells, sulfur-oxidizing bacteria, *E. aerogenes*, or recombinant *E. coli*) [11,12,13,14,15]. However, these enzymes, cells, and bacterial strains are sensitive to environmental changes; they are often limited to measuring dilute Cr(VI) concentrations and require deployment offsite under prepared controlled conditions [15].

The MFC is a self-sustaining device that oxidizes organic compounds in anaerobic anodes using electrogenic microbes, transporting electrons through an external circuit to the aerobic cathode, and converts biochemical energy into electrical energy [16]. Almost all studies in the MFC field are focused on electricity generation; however, continued efforts demonstrate increasing interest in making MFCs as biosensors for monitoring water quality parameters such as organic organics, heavy metals, biological oxygen demand (BOD), and volatile fatty acids [17]. MFCs involving Cr(VI) have been developed for different purposes. In the cathode of an MFC, Cr(VI) is used as an electron acceptor to be reduced or removed [18], to facilitate electricity production [19], or to detect trace Cr(VI) concentrations of 0.2–0.7 mg/L [20] in batch mode. In the anode of an MFC, Cr(VI) may be used as a toxic compound to cause the voltage to drop in the MFC because of the inhibition of anodic electrogenic bacteria activity. For example, Liu et al. developed a single-chambered MFC (SCMFC) sensor for monitoring Cr^6+^ shock (<10 mg/L) [21]. Similarly, Xu et al. developed a flat membrane-based MFC biosensor to monitor voltage changes that occur with toxic Cr^6+^ concentrations [22]. However, the calibration curve for the Cr^6+^ measurement was not established for these MFCs. Additionally, Cr(VI) is used as an electron acceptor in the anode to make the MFC an “actual” biosensor. Wang et al. inoculated *Ochrobactrum anthropi* YC152 into an MFC as an early warning device for accurately measuring Cr(VI) concentrations of 0.0125–5 mg/L in batch mode [10]. Wu et al. (2017) inoculated *E. aestuarii* YC211 into an MFC and discovered the operation performance of MFC-based biosensor was not affected by the surrounding environment [7]. MFC-based biosensors can accurately measure Cr(VI) concentrations of 2.5–60 mg/L in batch mode [7]. In practice, the concentrations of Cr(VI) emitted from various processes should be strictly controlled or continuously monitored for sustainable, clean, and green production. MFC-based biosensors inoculated with a single strain have shown higher selectivity and stability compared with those using bacterial consortia; however, such sensors narrow the Cr(VI) detection range [17].

A multistage MFC possesses unique attributes because the unconsumed substrate from the front MFC flows to the subsequent MFC, where the bacteria can continuously consume residual substrate [23]. Such system was applied as a BOD sensor, and results indicated that a wide range of BOD concentrations were obtained [23]. The present study developed a three-stage SCMFC biosensor to increase the range of Cr(VI) measurements in continuous mode. In this study, *E. aestuarii* YC211 was inoculated into the three-stage MFC system to evaluate its feasibility as a biosensor for in situ and real-time Cr(VI) measurements. Cr(VI) concentrations in leather processing wastewater were also measured using the developed system. To the best of our knowledge, this is the first report of continuous Cr(VI) measurement in leather processing wastewater using an MFC-based biosensor.

## 2. Materials and Methods

### 2.1. Bacterial Strains and Cultivation

The *E. aestuarii* YC211 inoculated to MFC were isolated by Wu et al. from the sludge of an electroplating wastewater treatment plant in New Taipei City, Taiwan [7]. A tryptic soy broth (TSB) supplemented with Na_2_Cr_2_O_7_, which together are called a TSBCr medium, containing 60 mg/L of Cr(VI) was used to cultivate the *E. aestuarii* YC211.

### 2.2. Construction of the SCMFC

An SCMFC was constructed to work as the Cr(VI) biosensor. The SCMFC comprised a 5 cm × 5 cm × 5 cm acrylic cube (working volume: 64 mL) with a surface area of 18 cm^2^, a graphite felt anode, and a Pt-free air cathode. The air cathode was made from carbon cloth (30 wt% PTFE, Fuel Cell Earth, Woburn, MA, SA), and a 50-μm microporous layer (MPL) was applied. The MPL was manufactured as previously described [24]. The anode and cathode were connected using an OK wire (silver plated copper wire) through a variable resistor.

The TSBCr medium (250 mL, 60 mg/L Cr(VI), 1/1000 TSB) containing 10^7.2^ cfu/mL *E. aestuarii* YC211 was placed in a sterile glass bottle and continuously recycled in the SCMFC at 30 °C with a 2000-Ω resistor using a submersible pump for cell immobilization under anoxic conditions for 10 d of liquid retention time (LRT) [7]. To maintain anoxic conditions, the feed solution (TSBCr medium) and SCMFC were purged with nitrogen gas before cell immobilization. Each SCMFC had an upper inlet port for medium and wastewater entrance and a lower port for medium and wastewater exit (Figure 1). Two small pores were located on the top of each SCMFC for online detection of pH, ORP, and DO. In this study, the experiment was conducted in two stages. The Cr(VI) concentration was either measured by an SCMFC in batch operation or by a three-stage SCMFC system in continuous-flow mode (Figure 1). The three-stage SCMFC system connected three single SCMFC in series and operated in continuous mode.

### 2.3. SCMFC in Batch Operation and Continuous-Flow Operation

A 1/1000 TSB medium containing 60 mg/L Cr(VI) was used as the anolyte to evaluate the performance of the SCMFC in batch operation. The circuit was adjusted using variable resistance (50–10,000 Ω) to obtain the relationships between the voltage output and current density and between the power density and current density of the SCMFC. In this study, the response time was set at 30 min for each resistance setting.

At the optimal operating resistance of the SCMFC, Cr(VI) with a final concentration (0.5–60 mg/L) was added to the 1/1000 TSB medium as the anolyte to obtain an appropriate response time or establish the relationship between the Cr(VI) concentration and voltage output of the SCMFC biosensor in batch operation. To examine the feasibility of the SCMFC and obtain the standard curve of Cr(VI) concentration versus voltage of the SCMFC in continuous mode, the anolyte containing various Cr(VI) concentrations was sequentially and continuously introduced to the SCMFC at 0.5–4 min LRT. The measurement data of the stable voltages of the SCMFC at different inlet Cr(VI) concentrations were used to create a standard curve.

### 2.4. Three-Stage SCMFC in Continuous-Flow Operation

The anolyte containing different Cr(VI) concentrations were continuously introduced to the three-stage SCMFC system at 2 min of LRT. The inlet Cr(VI) concentrations were divided into three levels. The 45 mg/L Cr(VI) medium was first introduced to the system. After 14 min, the 80 mg/L Cr(VI) medium was introduced; after 26 min, the 30 mg/L Cr(VI) was introduced. The estimated Cr(VI) concentration was calculated using the stable voltage in each SCMFC based on the established standard curve (described in Section 2.3); subsequently, the estimated Cr(VI) concentrations of water samples were obtained.

To evaluate the feasibility of the three-stage SCMFC system, actual tannery wastewater samples were collected. Wastewater samples A–H were obtained from the effluents of eight leather processing units. Cr(VI) concentrations in the actual tannery wastewater were measured using a three-stage SCMFC biosensor and a standard colorimetric method. Cr(VI) concentrations were measured by the biosensor in continuous-flow operation but by the standard colorimetric method in batch operation. The stable voltage for Cr(VI) measurement was recorded after 6.6 min of continuous operation. Two major water quality parameters (BOD_5_ and DO) of the wastewater affecting the voltage production of the SCMFC were determined. To understand the changes in the bacterial community of the SCMFC, the biofilm at the graphite felt was collected for bacterial community analysis through next-generation sequencing (NGS) before and after determining the Cr(VI) concentration in the tannery wastewater. All of the experiments were conducted using five separate SCMFCs or three groups of three-stage SCMFCs, and each analysis was conducted in triplicate.

### 2.5. Analysis

A specific Cr(VI) concentration was prepared from Na_2_Cr_2_O_7_ of analytical-grade chemicals through weighting and dissolution in water. The standard colorimetric method for Cr(VI) measurement was performed as described previously [3]. Briefly, the solution containing Cr(VI) was mixed with 0.25% *S*-diphenylcarbazide and 6 M H_2_SO_4_ and was determined at 540 nm using a UV–Vis spectrophotometer (Thermo Fisher Scientific Inc., Waltham, MA, USA). To perform the BOD_5_ analysis, the standard BOD method 5210 B was adopted. ORP, pH, and DO were measured using a multiparameter portable meter 3630 IDS (Xylem Analytics, Beverly, MA, USA).

To understand the changes in the bacterial community of the three-stage SCMFC system, the biofilm at the anode was collected and analyzed. Bacterial DNA was extracted using a Fast DNA Spin Kit (MP Biomedicals, Santa Ana, CA, USA). Polymerase chain reaction (PCR) was performed to amplify the V3–V4 region of the eubacterial 16S ribosomal RNA fragments, and PCR profiling was performed as described previously [7]. The PCR-amplified 16S rRNA gene fragments were purified and preprocessed based on the methods described by Naz et al. (2016) [25]. The 16S rRNA gene sequence data were analyzed using QIIME software (version 1.17). After processing, the qualified reads were clustered into operational taxonomic units at a 97% sequence similarity through the UCLUST method [26]. Taxonomic assignment was performed on representative sequences using the RPD classifier.

The SCMFC voltage was measured using a multimeter (Model 2700, Keithley Instruments, Inc., Solon, OH, USA) and recorded by a personal computer through a data acquisition system (Testpoint, Capital Equipment Co., Richmond, VA, USA). The current (*I*, amp) was obtained by dividing the resistance (*R*, ohm) by the measured voltage (*V*, volt). The power (*P*) was calculated as *P* (watt) = *I* (amp) × *V* (volt). Power density and current density were measured as watts and amperes per unit of the total surface area of the anode.

## 3. Results and Discussion

### 3.1. Effect of External Resistance on SCMFC Performance in Batch Operation

When the potential of the SCMFC reached a steady state (after 19–21 days of immobilization operation), the biofilm in the anode was considered stable or mature [3]. To optimize the SCMFC biosensor’s signal and performance, the effects of external resistance on the biosensor in batch operation were first evaluated. Figure 2 presents the curves of polarization and power density obtained in an SCMFC biosensor during the stable phase of power generation. Results revealed that the voltage of the SCMFC decreased with increasing current density and exhibited a typical polarization curve [27]. A maximum voltage of 926 ± 32.5 mV occurred at 10,000 Ω. For voltage output to stabilize, 20–30 min was required. In addition, the power density of the SCMFC initially increased with current density but started to decrease after a certain point. The maximum power density was 167.5 ± 6.8 mW/m^2^; the voltage and external resistance were 367.2 ± 42.5 mV and 500 Ω, respectively, under such conditions. Therefore, the external resistance for each MFC within the SCMFC was set at 500 Ω for subsequent experiments [27]. The maximum power density of the SCMFC is superior to that reported in earlier studies (100.1 ± 1.2 mW/m^2^) using a dual-chambered MFC inoculated with the same strain [7].

### 3.2. Effect of Cr(VI) Concentration on SCMFC Performance in Batch Operation

Wu et al. (2017) demonstrated that the performance of a dual-chambered MFC inoculated with *E. aestuarii* YC211 was not notably affected by water quality measurements (coexisting ions, pH, or NaCl concentration) [7]. Therefore, the effect of Cr(VI) concentration on SCMFC performance in batch operation was evaluated. Under optimal operating conditions, the relationship between the Cr(VI) concentration and voltage output of the SCMFC was characterized. In this study, a response time of 20 min was required to obtain a stable voltage at 500 Ω. Figure 3 indicates that a negative correlation was observed for Cr(VI) concentrations ranging from 0.5 to 60 mg/L. When the Cr(VI) concentration was lower than 0.5 mg/L or higher than 60 mg/L, no linear relationship was observed. The regression equation for Cr(VI) concentration and voltage output of the SCMFC biosensor was determined to be *y* = −5.7668*x* + 656.09 (*r*^2^ = 0.9997). Wu et al. (2017) using a dual-chambered MFC inoculated with YC211 also observed a linear relationship (*y* = −2.3256*x* + 517.15), but their Cr(VI) concentration ranges narrowed down to 2.5–60 mg/L. The higher slope or voltage drop that occurred in our SCMFC suggests the biosensor exhibits some competitive advantages compared with the dual-chambered MFC because of the SCMFC’s relative sensitivity and wide measurement range.

### 3.3. Effect of Flowrate and Cr(VI) Concentration on the Stable Time of SCMFC Performance in Continuous-Flow Operation

Based on practical application, an SCMFC biosensor should be developed for the in situ or real-time measurement of a wide range of Cr(VI) concentrations. Thus, the biosensor should be in continuous-flow operation. Figure 4A indicates that the stable time for the potential production of the SCMFC shortens with an increasing retention time. A fast flowrate resulted in an evidently long stable time. The 2-min LRT achieved a relatively stable time (322 s) for potential production of the SCMFC. Thus, the retention time of the SCMFC was set at 2 min in continuous-flow operation during the subsequent experiment. Figure 4B illustrates the relationship between Cr(VI) concentration and stable time at 2 min of LRT. Results indicated that the stable time for potential production of the SCMFC increased with Cr(VI) concentration in continuous-flow operation. For 5–30 mg/L inlet Cr(VI) concentrations, 198–400 s of stable time was required. When Cr(VI) concentration was lower than 5 mg/L or higher than 30 mg/L, the linear relationship did not apply.

### 3.4. Establishment of a Standard Curve for Determination of Cr(VI) Concentration by the SCMFC in Continuous-Flow Operation

To establish the standard curve for determination of Cr(VI) concentration by the SCMFC, different Cr(VI) concentrations were sequentially fed to the SCMFC at 2-min LRTs in continuous-flow operation. For the initial 3 min, the anolyte was fed into the SCMFC, and the voltage output of the SCMFC reached 728 ± 16.2 mV. After 3 min, the anolyte containing 5 mg/L Cr(VI) was fed, and the voltage output gradually decreased and stabilized at 645 ± 9.2 mV from minutes 7–11. After 11 min, the voltage gradually recovered to 721 ± 21.6 mV. When the anolyte containing 10 mg/L Cr(VI) was fed, the voltage gradually decreased and stabilized at 624 ± 6.1 mV during minutes 16–19. After 19 min, the anolyte containing 15 mg/L Cr(VI) was fed, and the voltage output gradually decreased and stabilized at 602 ± 5.1 mV during minutes 25–27. A similar variation and tendency was observed after subsequent tests (Figure 5A). When 30 mg/L Cr(VI) was fed, the voltage output steadily decreased to 537 ± 4.6 mV. In this study, the stable times for various Cr(VI) concentrations were 3.3–6.6 min.

Figure 5B shows the standard curve for the determination of Cr(VI) concentration by the SCMFC with 2-min LRT in continuous-flow operation. A strong negative correlation was observed between Cr(VI) concentrations and voltage output. The regression equation was determined to be *y* (voltage, mV) = −4.2629*x* [Cr(VI) concentration, mg/L] + 666.27 (*r*^2^ = 0.9994) when Cr(VI) concentration ranged from 5–30 mg/L. Thus, using the standard curve, the Cr(VI) concentration in the wastewater can be determined in 6.6 min using an SCMFC biosensor in continuous-flow operation. Compared with the previous results of batch operation (Section 3.2), the range measurement of Cr(VI) concentration by the SCMFC is narrower in continuous-flow operation. Currently, no published studies exist regarding the continuous Cr(VI) measurement by MFC biosensors. Compared with previous studies using different MFC biosensors in batch operation, the measuring ranges of 2.5–60 mg/L by Wu et al. [7], 0.0125–5 mg/L by Wang et al. [10], 0.2–0.7 mg/L by Zhao et al. [20], 1–8 mg/L by Liu et al. [21], and 5–20 mg/L by Xu et al. [22] strongly suggest that the SCMFC biosensor has considerable potential for Cr(VI) measurement because of its wide dynamic range and continuous measurement.

### 3.5. Cr(VI) Measurement of Artificial Tannery Wastewater Using Three-Stage SCMFC Biosensor in Continuous-Flow Operation

The SCMFC biosensor could accurately determine 5–30 mg/L of Cr(VI) at 2-min LRT in continuous-flow operation. To expand the Cr(VI) measurement range, a three-stage SCMFC biosensor or system (Figure 1) was developed to determine the Cr(VI) concentration from artificial and actual tannery wastewater. Figure 6 presents the effect of Cr(VI) concentrations sequentially fed to the system on the voltage output of the SCMFC. In the first stage (through 14 min), the initial voltage in MFC 1 was 725 ± 10.5 mV; it gradually decreased and stabilized at 536.5 ± 2.8 mV between 6.6 and 10 min while 45 mg/L Cr(VI) was introduced. The voltage output (536.5 mV) of MFC 1 converted to Cr(VI) concentration in artificial tannery wastewater was 30.44 mg/L according to the regression equation presented in Figure 5B. After 3 min of operation, MFC 2 received the wastewater from MFC 1, and the voltage in MFC 2 gradually decreased before leveling off at 606.5 ± 5.1 mV during minutes 7–13. The Cr(VI) concentration in the artificial tannery wastewater was calculated as 14.02 mg/L. After 4 min, the MFC 2 effluent flowed into MFC 3; the lack of voltage change indicated a zero Cr(VI) concentration. Therefore, the inflow Cr(VI) concentration was reduced in the three-stage SCMFC biosensor, and the value was 44.46 (30.44 + 14.02) mg/L.

In the second stage (14th–26th min), the voltage (726 ± 12.8 mV) in MFC 1 gradually decreased and stabilized at 537.3 ± 1.4 mV during minutes 20–22, while 80 mg/L Cr(VI) was introduced. The Cr(VI) concentration in artificial tannery wastewater was calculated as 30.25 mg/L. MFC 2 received the MFC 1 effluent, and the voltage in MFC2 gradually decreased before leveling off at 539.0 ± 1.2 mV during minutes 20–22. The Cr(VI) concentration in artificial tannery wastewater was calculated as 29.86 mg/L. Finally, the MFC 2 effluent flowed into MFC 3, and the voltage in MFC 3 stabilized at 580.0 ± 5.3 mV during minutes 20–25. The Cr(VI) concentration in wastewater was calculated as 20.24 mg/L. Thus, the inflow Cr(VI) concentration was reduced in the three-stage SCMFC biosensor, and its value was 80.35 (30.25 + 29.86 + 20.24) mg/L.

In the third stage (26th–35th min), the voltage in MFC 1 first rose to 718 ± 18.2 mV and stabilized at 538.5 ± 0.6 mV during minutes 32–35 while 30 mg/L Cr(VI) was introduced. The Cr(VI) concentration in wastewater was calculated as 29.97 mg/L. After 30 min of operation, the voltages in MFC 2 and MFC 3 remained at 725.6 ± 13.4 mV, suggesting that Cr(VI) did not exist in the wastewater. Therefore, the inflow Cr(VI) concentration was reduced in the three-stage SCMFC biosensor, and its value was 29.97 mg/L. The results indicated deviations were –1.2%, 0.44%, and –0.09% for determining 45, 80, and 30 mg/L Cr(VI), respectively, using the three-stage SCMFC biosensor.

### 3.6. Cr(VI) Measurement of Actual Tannery Wastewater Using Three-Stage SCMFC Biosensor and Bacterial Community in Three-Stage SCMFC Biosensor in Continuous-Flow Operation

The three-stage SCMFC biosensor exhibited low deviations in the range of −1.2% to 0.44% in the Cr(VI) measurement from artificial tannery wastewater. However, the concentration and composition of organic compounds vary in actual tannery wastewater; therefore, the feasibility of the three-stage SCMFC biosensor for Cr(VI) measurement from an actual tannery wastewater should be evaluated. A previous study revealed that water quality parameters did not significantly affect the performance of an MFC inoculated with *E. aestuarii* YC211 [7]. Thus, this study focuses on the effect of crucial electron donors (e.g., organic compounds) and acceptors (e.g., Cr^6+^, O_2_) existing in the tannery wastewater. The mechanism for Cr(VI) measurement using the SCMFC biosensor inoculated with *E. aestuarii* YC211 is possibly as follows [10]:Anode: Organics → CO_2_ (or other intermediates) + H^+^ + e^−^ (by *E. aestuarii* YC211)(1)
Cr^6+^ + e^−^ → Cr^3+^ (by *E. aestuarii* YC211)(2)
O_2_ + H^+^ + e^−^ → H_2_O (by chemical reaction)(3)
Cathode: O_2_ + H^+^ + e^−^ → H_2_O (by chemical reaction)(4)

The higher the organic concentrations entering the SCMFC are, the greater the voltages are produced. The higher the Cr^6+^ or O_2_ concentrations are in the anode, the fewer electrons are transferred to the cathode, and the fewer voltages are produced. Thus, the potential output will decrease with increasing Cr^6+^ concentration except in a case of O_2_ interference. According to the result in Figure 5B, the SCMFC biosensor for Cr(VI) measurement range is 5–30 mg/L; therefore, the theoretical Cr(VI) measurement range of the three-stage SCMFC biosensor would be extended to 5–90 mg/L by cumulating its voltage [23]. Table 1 lists Cr(VI) measurements of the effluents of eight leather processing units (A–H) by three-stage SCMFC biosensor in continuous-flow operation and using colorimetric method in batch operation. Results indicated a higher deviation (>12%) in the Cr(VI) measurement of effluents of units A and H (2.6 and 124.5 mg/L) by our system compared with the colorimetric method. This can be explained by the fact that the measurement range was not within the optimal 5–90 mg/L measurement range. A lower deviation (<7%) was observed in Cr(VI) measurement from effluents of units B–G (6.8–84.2 mg/L) by our system compared with the colorimetric method because these units’ concentrations fell in the measurement range. Compared with the deviation (−1.2%–0.44%) of Cr(VI) measurement of artificial wastewater using a similar system (Figure 6), Cr(VI) measurement of actual wastewater is less accurate, mainly because of the effects of BOD and DO in the wastewater [28].

If the inlet organic concentration of BOD in the wastewater was >256 mg/L (the anolyte as BOD), the initial potential of the three-stage SCMFC system would be higher than the expected value. Although the amplitude of the voltage drop of the SCMFC was identical at the same Cr(VI) concentration, the stable voltage was higher than the theoretical voltage. Thus, the Cr(VI) concentration was underestimated according to the standard curve and resulted in a negative deviation (for units B, C, D, and E). Conversely, BOD in the wastewater was <256 mg/L, which resulted in a lower initial potential than the expected value. At the same potential drop, the voltage value at the equilibrium would be lower than the theoretical voltage. Thus, the Cr(VI) concentration was overestimated and resulted in a positive deviation (units F and G). In addition, high DO content competes with Cr(VI) as an electron acceptor and causes the potential drop of the non-Cr(VI) factor. This resulted in the Cr(VI) concentration being overestimated with a positive deviation (unit F). However, a combination effect (DO and organics) is possible in this system [28].

To understand the changes in the bacterial community of the three-stage SCMFC biosensor after operation, the biofilm at the anode was analyzed through NGS [29]. Figure 7 indicates the relative abundances of the bacterial 16S rRNA gene sequences in three separate SCMFCs (MFC 1, MFC 2, and MFC 3) after measuring the Cr(VI) concentration from the effluents of eight leather processing units. At first, *E. aestuarii* YC211 alone was inoculated in the three-stage SCMFC. After the SCMFC system was used to treat the wastewater, the bacterial community in MFC 1 was the simplest, and only four strains were observed: *E. aestuarii* (97.5%), *O. anthropi* (1.45%), *Exiguobacterium* sp. (0.92%), and *O. tritici* (0.13%). These four strains are classified as Cr(VI)-reducing bacteria [7,10,30,31]; thus, they can endure relatively high Cr(VI) concentrations. The complexity of the bacterial community in MFC 2 was the second simplest, and the community increased to seven strains. The strains included *Lysinibacillus fusiformis*, *Pseudomonas putida*, and *Arthrobacter* sp. in addition to the four strains that existed in MFC 1. The Cr(VI)-reducing capability of these three strains has been reported [32,33,34]. In MFC2, the dominant strain remained *E. aestuarii* (95.12%), and other strains were represented at <1.5%. The bacterial community in MFC 3 was the most complicated because the inlet Cr(VI) concentration was the lowest. The bacterial community increased to 10 strains. The strains comprised *Microbacterium* sp., *Streptomyces coelicolor*, and *Staphylococcus aureus* and the seven strains that existed in MFC 2. These three new strains have been regarded as potential Cr(VI)-reducing bacteria [4,35,36]. In MFC 3, the dominant strain was *E. aestuarii* (92.54%), and other strains were represented at <3.0%. The results of the aforementioned studies demonstrate that reliable three-stage SCMFC performance (<7% deviation) should be attributed to a stable bacterial community (>92.5% original strains) present during the treatment period (four months).

## 4. Conclusions

In this study, a three-stage SCMFC biosensor inoculated with *E. aestuarii* YC211 was developed for the continuous measurement of Cr(VI) from actual tannery wastewater. The system exhibits competitive advantages over previous MFCs, including a simpler structure, higher accuracy, shorter measurement time, and wider measurement range. Through NGS analysis, the original inoculated *E. aestuarii* remained dominant (92.5%–97.5% of the total bacterial community) in the three-stage SCMFC even after treating the actual tannery wastewater. The parameters most affecting the accurate Cr(VI) measurement of the system were the concentrations of organics and oxygen in the wastewater. Although both these unexpected factors may slightly restrict the system’s application, the system has a potential as an early warning device with wide dynamic range for in situ, real-time, and continuous Cr(VI) measurement. Moreover, the three-stage SCMFC can further expand to a multistage MFC biosensor by connecting several separate SCMFCs in series and increase its application fields or range.

## Figures and Tables

**Figure 1 sensors-19-01418-f001:**
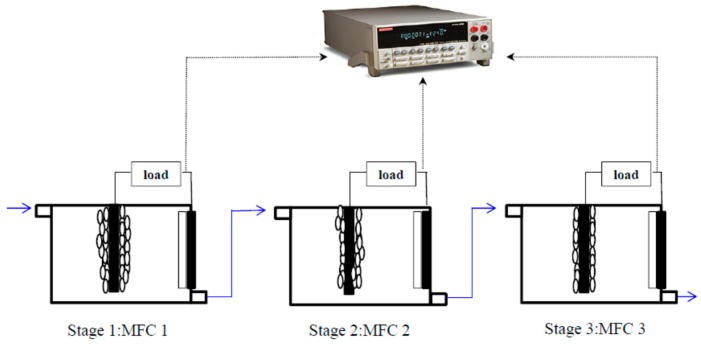
Schematic configuration of three-stage SCMFC system.

**Figure 2 sensors-19-01418-f002:**
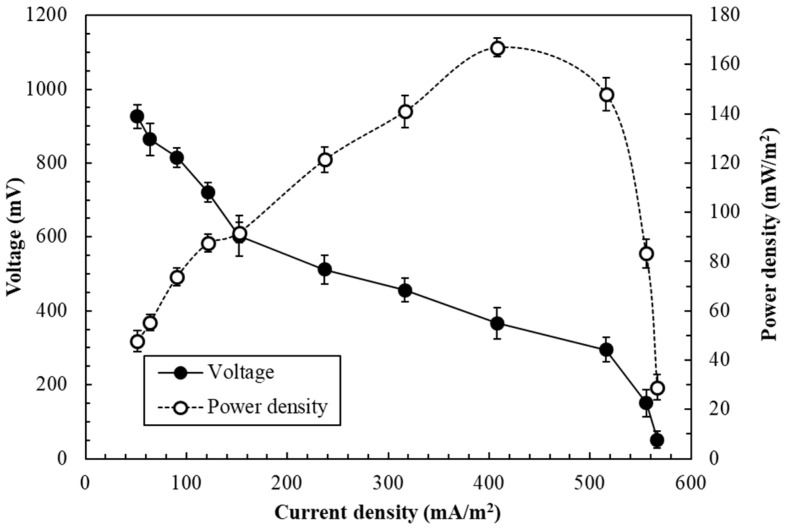
Curves of polarization and power density obtained in an SCMFC biosensor inoculated with *E. aestuarii* YC211 in the batch operation (anolyte: 1/1000 TSB supplemented with 60 mg/L Cr(VI), external resistance: 50–10,000 Ω).

**Figure 3 sensors-19-01418-f003:**
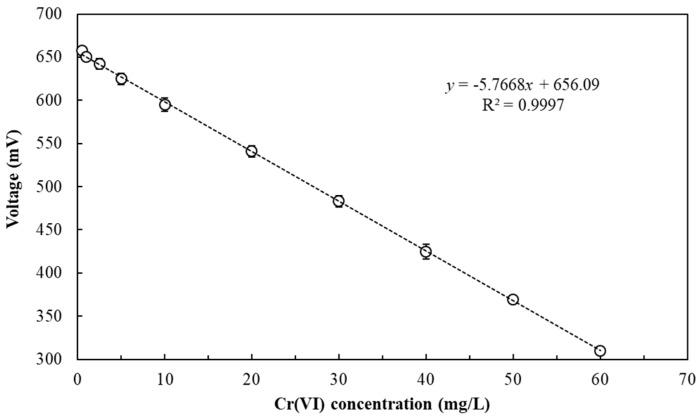
Relationship between Cr(VI) concentration and voltage output of the SCMFC biosensor inoculated with *E. aestuarii* YC211 in the batch operation (anolyte: 1/1000 TSB supplemented with different Cr(VI) concentrations, response time: 20 min, external resistance: 500 Ω).

**Figure 4 sensors-19-01418-f004:**
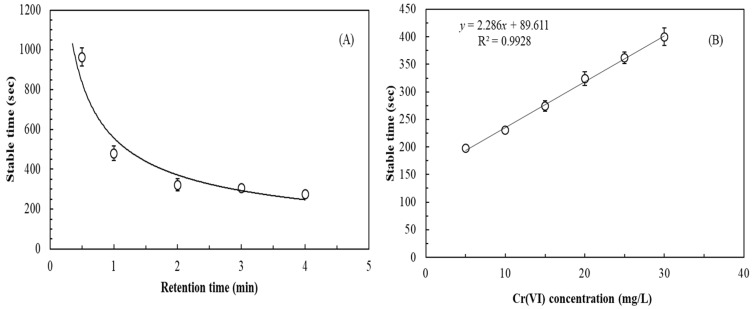
Effects of (**A**) retention time (anolyte: 1/1000 TSB supplemented with 20 mg/L Cr(VI), external resistance: 500 Ω); (**B**) Cr(VI) concentration (anolyte: 1/1000 TSB supplemented with different Cr(VI) concentrations, liquid retention time: 2 min, external resistance: 500 Ω) on the stable time of the SCMFC biosensor inoculated with *E. aestuarii* YC211 in the continuous-flow operation.

**Figure 5 sensors-19-01418-f005:**
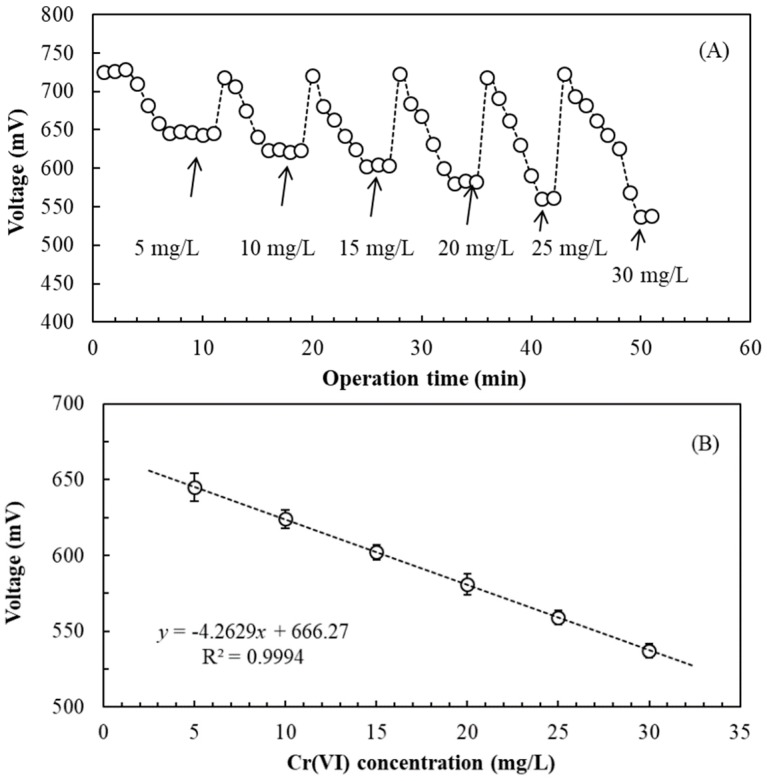
(**A**) Effect of different inlet Cr(VI) concentrations on voltage output of the SCMFC biosensor inoculated with *E. aestuarii* YC211 in the continuous-flow operation (liquid retention time: 2 min, external resistance: 500 Ω); (**B**) Relationship between Cr(VI) concentration and voltage output of the SCMFC biosensor inoculated with *E. aestuarii* YC211 in the continuous-flow operation (voltage data obtained by 6.6-min stable time).

**Figure 6 sensors-19-01418-f006:**
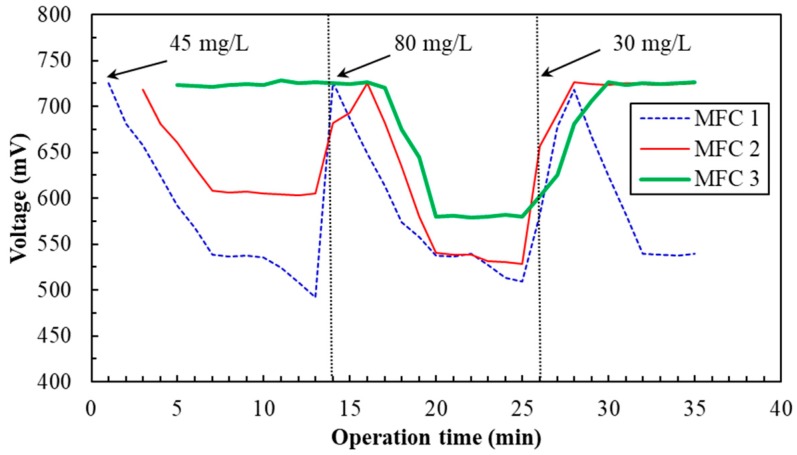
Effect of Cr(VI) concentrations sequentially fed to the system on the voltage output of the three-stage SCMFC biosensor inoculated with *E. aestuarii* YC211 in continuous-flow operation (liquid retention time: 2 min, stable time: 6.6 min, external resistance: 500 Ω).

**Figure 7 sensors-19-01418-f007:**
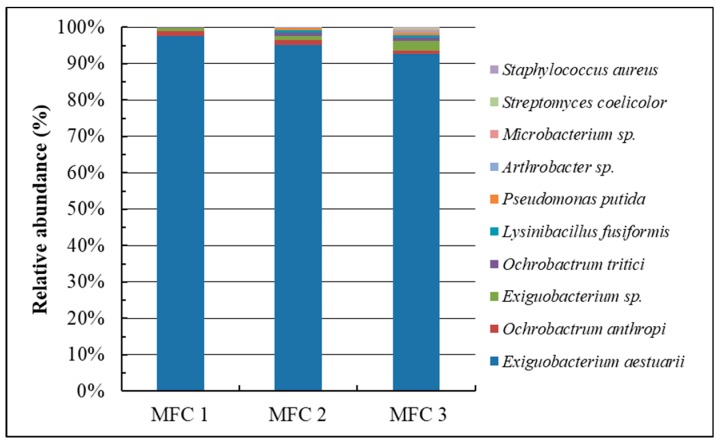
Relative abundances of the bacterial 16S rRNA gene sequences in the biofilm of three-stage SCMFC biosensor after measuring the Cr(VI) concentration from the effluents of eight leather processing units.

**Table 1 sensors-19-01418-t001:** Cr(VI) measurement from effluents of eight leather processing units by three-stage SCMFC biosensor in continuous-flow operation and colorimetric method in batch operation.

Tannery Wastewater								
Unit	A	B	C	D	E	F	G	H
BOD	250	350	360	420	460	180	230	560
DO	2.1	1.8	2.3	2.6	2.5	3.1	1.6	1.2
MFC biosensor	2.6 ± 0.08	6.8 ± 0.21	12.8 ± 0.51	32.4 ± 2.01	52.3 ± 1.62	75.6 ± 2.41	84.2 ± 3.64	124.5 ± 7.26
Colorimetric method	2.3 ± 0.04	7.1 ± 0.13	13.6 ± 0.35	33.6 ± 1.06	56.1 ± 1.81	71.2 ± 2.56	82.6 ± 4.06	142.1 ± 6.04
Deviation (%) *	13.0%	−4.2%	−5.9%	−3.6%	−6.8%	6.2%	1.9%	−12.4%

* The determined value by SCMFC biosensor compared to that by colorimetric method.

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
