# Peer review of "Three-Stage Single-Chambered Microbial Fuel Cell Biosensor Inoculated with *Exiguobacterium aestuarii* YC211 for Continuous Chromium (VI) Measurement"

_sensors, 2019, doi:10.3390/s19061418_

Round 1

Reviewer 1 Report

This research article is the investigation of the availability of Microbial Fuel Cell as a biosensor for Cr(VI). In this system, the cell voltage is decreased by inhibition of the activity of Exiguobacterium aestuarii by Cr(VI). This phenomenon depends on Cr(IV) concentration, so actual Cr(VI) concentration can calculate by the relationship between the voltage and reference Cr(VI) concentration. Is it right? This system still has several challenges for practical use, it is very interesting and important. The worth of this article is sufficient for publication. However, I have a few questions.

Comments

(1) I think that the Cr(IV) in the wastewater is completely oxidized during the three-stage MFC treatment, is it correct? Where does this oxidation reaction occur? (anode surface, liquid phase or other) I recommend that the author add more information about the mechanisms of Cr(IV) oxidation with MFC.

(2) How does voltage inhibition occur? Generally speaking, the more poison concentration increases, the more inhibition against the microbial activity occurs. However, in this study, the inhibition effect against MFC became a constant value when Cr(VI) concentration is above 30 mg/L (fig. 6, 80 mg/L). Why is it?

Author Response

In this system, the cell voltage is “not” decreased by inhibition of the activity of Exiguobacterium aestuarii by Cr(VI). E. aestuarii YC211 is a Cr(VI)-reducing bacteria which can efficiently remove Cr(VI). The mechanism for Cr(VI) measurement using the SCMFC biosensor inoculated with E. aestuarii YC211 is described as lines 314-321. In this study, Cr6+ played an electron acceptor in the anode. The higher the Cr6+ concentrations in the anode, the fewer electrons are transferred to the cathode, as measured the fewer voltages generated in the experiment. Thus, the potential output will decrease with increasing Cr6+ concentration. Thank for reviewer’s kind suggestion.

respond to comment:

(1)The Cr(IV) in the wastewater is completely reduced to Cr(III) during the three-stage MFC treatment by E. aestuarii YC211 in the anode rather than oxidation. The mechanism for Cr(VI) measurement using the SCMFC biosensor inoculated with E. aestuarii YC211 is described as lines 314-321. Thank for reviewer’s kind suggestion.

(2)E. aestuarii YC211 is a Cr(VI)-reducing bacteria and can efficiently remove Cr(VI). Thus, 80 mg/L of Cr6+ is not toxic to E. aestuarii YC211. The Cr(IV) in the wastewater is completely “reduced” by E. aestuarii YC211 in the anode during the three-stage MFC treatment. The mechanism for Cr6+ measurement using the SCMFC biosensor inoculated with E. aestuarii YC211 is described as lines 314-321. In this study, Cr6+ played an electron acceptor in the anode. The higher the Cr6+ concentrations in the anode, the fewer electrons are transferred to the cathode, as measured by the fewer voltages generated in the experiment. Thus, the potential output will decrease with increasing Cr6+ concentration. Eventually, Cr(IV) is completely reduced by E. aestuarii YC211 and stabilizes.

The inlet Cr(IV) concentrations were introduced into the system in three stages. The 45 mg/L Cr(IV) medium was first introduced. After 14 min, the 80 mg/L Cr(IV) medium was introduced; 26 min later, the 30 mg/L Cr(IV) was introduced. Because 45 mg/L Cr(IV) was completely reduced by E. aestuarii YC211 in MFC 1 and MFC 2 , none of Cr(IV) is measured in MFC 3 with a constant voltage observed after first introduced inlet. Green line (fig. 6) indicates the voltage of MFC 3. Thank for reviewer’s kind suggestion.

Reviewer 2 Report

In this study, a three-stage single-chambered MFC (SCMFC) biosensor was developed to investigate its Chromium(VI) sensing potential. Real tannery wastewater was used as well as synthetic wastewater in order to measure Cr+6 detection range. The experiments were planned and conducted soundly, interpretation of results is logical. Presentation of results is also nicely done. Reviewer recommends to accept the manuscript to be published if the following questions are answered satisfactorily.

General question: How practical is it to build in situ, real-time and continuous Cr+6 MFC biosensor for dealing with real wastewater, using anodes inoculated with single species? As you showed in your results, microbial culture changes over time. What kind of maintenance, or replacement frequency, pre-treatment of samples do you think will be needed for bring this technology to a commercial level?     

Line 114: What is OK wire?

Lines 125-126: Were three MFCs electrically connected in series? Or were they placed in series to share the feedstock flow? A detailed description is needed. Figure 1 is a bit misleading. Please amend Fig.1 to show electrical connection clearly.

Line 127, Figure 1: Some part of the figure is missing.

Line 203-204: Was the external resistance of 500 Ω for the entire SCMFC (3 MFCs connected in series) or 500 Ω for each MFC within the SCMFC?

Lines 292-293: Cr+6 accumulation? Do you mean reduction? Accumulation implies chromium stayed in MFCs and did not leave the system. Is that case, how much Cr+6 can be accumulated in SCMFC biosensor before reaching to a threshold? Wouldn’t this affect for repeated operation/measurement? It was continuous system. How was Cr+6 accumulated?   

Lines 303-310: Same as above.

Figures 5, 6, 7: Why do they have green background?

Lines 390-391: How long was the treatment period? Please specify actual time spent for this treatment.

Author Response

(1)Thank for reviewer’s appreciation.

(2)Thank for reviewer’s comment. It is good question for practical challenge to meet unknown wastewater sample. After treating the actual tannery wastewater with the SCMFC, the original inoculated E. aestuarii remained dominant (>92.5%). Although the system inoculated with single Cr(VI)-reducing strain is function well in continuous Cr(VI) measurement of tannery wastewater, the other wastewater containing Cr(VI) remain further investigation. Thus, we claim “the three-stage SCMFC biosensor has potential as an early warning device with wide dynamic range for in situ, real-time, and continuous Cr(VI) measurement of tannery wastewater.” in lines 29-31. In all, the systems have measured Cr(VI) concentration from tannery wastewater for 2 years without additional maintenance or treatment.

(3)OK wire is a silver-plated copper wire, which has been added in lines 110-111.

(4)The Fig.1 of electrical connection in this study has been properly modified. In fact, the voltage was accumulated from separate SCMFC biosensor to calculate the inflow Cr(VI) concentration in the wastewater. Thank for reviewer’s kind suggestion.

(5)The Fig.1 in this study has been properly modified and supplemented the missing part. Thank for reviewer’s kind suggestion.

(6)The sentence of “Therefore, the external resistance was set at 500 Ω for subsequent experiments” has been amended to “Therefore, the external resistance for each MFC within the SCMFC was set at 500 Ω for subsequent experiments” (lines 195-196). Thank for reviewer’s kind suggestion.

(7)The phenomenon of Cr6+ in the system is indeed “reduction” rather than “accumulation”. The word of “accumulated” has been amended to “reduced” in line 285, 293, 299. Thank for reviewer’s comment.

(8)The phenomenon of Cr6+ in the system is indeed “reduction” rather than “accumulation”. The word of “accumulated” has been amended to “reduced” in line 285, 293, 299. Thank for reviewer’s comment. 

(9)None of the green background in figs 5-7 has been found. We speculate that it may be a typesetting problem.

(10)The authors have supplemented the treatment period (four months) in line 382 as reviewer’s suggestion.

Reviewer 3 Report

This work studied the possibility of building a biosensor of Cr(IV) concentration in solution using a cascade of three single chamber microbial fuel cells.

The assumption is that in a selected inoculum of Cr(IV) reducing bacteria, enriched with BOD, the addition of a flow of Cr(IV) causes the decay of the generated current in the MFC, proportional to the Cr(IV) concentration. It is because Cr(IV) compete with the anode in collecting electrons from BOD microbial degradation.

The concept was already studied for some extend in previous works, the main novelty is in using a cascade of 3 subsequent MFCs in order to increase the sensitivity of the biosensor for Cr(IV) concentration.

An inverse liner proportionality between the MFC voltage and the Cr(IV) concentration is shown.

The data reported are of interest, including the results of sequencing (Cr(IV) degradation bacteria pool) and deserve publication.

The main issue is the English and in the description of the results, that sometime is a little confusing.

In particular, these minor notes should be addressed:

- The caption of Figure 5 and 6 are not clearly exhaustive. The rows in Figure 5A (and 6) seem shifted on the right. Also the red frame of Figure 6 meaning is not evident.

- It is not actually clear how many MFCs have been operated in total , and at the same retention time. (replicates) al least.

Figure 1: figure is cut on the right

 It should be better to refer to “anolyte” instead “1/1000 TSB media” after the first time.

 Please simplify the sentences avoiding repetition, for instance in the following sentence:

Line 137: “To examine the feasibility of the SCMFC in continuous mode, the 1/1000 TSB medium containing 20 mg/L Cr(VI) was continuously introduced to the SCMFC at 0.5–4 min LRT. To obtain an appropriate response or stable time of the SCMFC in continuous mode, the 1/1000 TSB medium containing 5–60 mg/L of Cr(VI) was continuously introduced to the SCMFC with a 2-minLRT.To obtain the standard curve of Cr(VI) concentration versus voltage of the SCMFC, 1/1000 TSB media containing various Cr(VI)concentrations (0, 5, 10, 15, 20, 25, and 30 mg/L) were sequentially and continuously introduced to the SCMFC with 2 min of LRT for five cycles.”

 Line 332: “by accumulating its voltage” means “cumulating its voltage”?

 Line 204:please report the max power density, not the voltage, otherwise please correct the sentence (100.1 ± 1.2 mV)

Author Response

(1)Thank for reviewer’s appreciation.

(2)Authors have amended the caption of Figure 5 and 6 and the red frame of Figure 6 was removed as reviewer’s suggestion.

(3)All the experiments were conducted using five separate SCMFCs or three groups of three-stage SCMFCs, and each analysis was conducted in triplicate (lines 156-158). Thank for reviewer’s kind suggestion.

(4)Authors have amended the Fig.1 to show other parts. Thank for reviewer’s kind suggestion.

(5)Authors have amended “1/1000 TSB media” to “anolyte” (line 134, 139, 241, 242, 244, 246, 335) as reviewer’s kind suggestion.

(6)Thank for reviewer’s kind suggestion. Author has amended the following sentences avoiding repetition. To examine the feasibility of the SCMFC and obtain the standard curve of Cr(VI) concentration versus voltage of the SCMFC in continuous mode, the anolyte containing various Cr(VI) concentrations was sequentially and continuously introduced to the SCMFC at 0.5–4 min LRT (lines 133-136).

(7)Thank for reviewer’s kind suggestion. Authors have amended “accumulating” to “cumulating” (line 324).

(8)Thank for reviewer’s kind suggestion. Authors have amended “100.1 ± 1.2 mV” to “100.1 ± 1.2 mW/m2” (line 197).